# Multi-Modal Sarcasm Detection with Sentiment Word Embedding

**Hao Fu [1], Hao Liu [1], Hongling Wang [2], Linyan Xu [1] , Jiali Lin [3,*] and Dazhi Jiang [1,*]**

1   Department of Computer Science, Shantou University, Shantou 515063, China; 21hfu@stu.edu.cn (H.F.); 20hliu2@alumni.stu.edu.cn (H.L.); linyanxu@stu.edu.cn (L.X.)
2   Shenzhen Tobo Software Co., Ltd., Shenzhen 518055, China; wanghongling@tobosoft.com.cn
3   Bussiness School, Shantou University, Shantou 515063, China
*   Correspondence: jllin@stu.edu.cn (J.L.); dzjiang@stu.edu.cn (D.J.)

**Abstract:** Sarcasm poses a significant challenge for detection due to its unique linguistic phenomenon where the intended meaning is often opposite of the literal expression. Current sarcasm detection technology primarily utilizes multi-modal processing, but the connotative semantic information provided by the modality itself is limited. It is a challenge to mine the semantic information contained in the combination of sarcasm samples and external commonsense knowledge. Furthermore, as the essence of sarcasm detection lies in measuring emotional inconsistency, the rich semantic information may introduce excessive noise to inconsistency measurement. To mitigate these limitations, we propose a hierarchical framework in this paper. Specifically, to enrich the semantic information of each modality, our approach uses sentiment dictionaries to obtain the sentiment vectors by evaluating the words extracted from various modalities, and then combines them with each modality. Furthermore, in order to mine the joint semantic information implied in the modalities and improve measurement of emotional inconsistency, the emotional information representation obtained by fusing each modality's data is concatenated with the sentiment vector. Then, cross-modal fusion is performed through cross-attention, and, finally, the sarcasm is recognized by fusing low-level information in the cross-modal fusion layer. Our model is evaluated on a public multi-modal sarcasm detection dataset based on Twitter, and the results demonstrate its superiority.

**Keywords:** multi-modal; emotional inconsistency; sentiment vector

## 1. Introduction

Sarcasm is a unique form of emotional expression where individuals express contempt that contradicts their actual emotions or intentions [1]. The prevalence of multi-modal messages rich in sarcasm on social media has made detecting sarcasm in such expressions a new and challenging research problem in sentiment analysis [2]. Generally, intentional ambiguity often characterizes sarcasm, which makes the detection process a challenge, particularly in multi-modal expressions that use text to describe images. For instance, Figure 1 depicts a diagram, that includes the accompanying text "158 new people a day in # austin # welcome". The text alone seems to suggest that the author is welcoming the 158 newcomers, but the countless vehicles in the image indicate that the author is actually satirizing the heavy traffic.

Traditional sarcasm detection methods are generally based on rules and statistical knowledge, which require manual extraction of specific vocabulary and punctuation as features. However, these methods necessitate professional domain knowledge and lack robustness, which means that these methods require the design of different rules and different feature extraction methods for different scenarios, resulting in high costs. Given the widespread effectiveness of deep learning methods, they are also used to automatically acquire useful features for sarcasm detection. For instance, Gupta et al. [3] proposed an

architecture using the RoBERTa model that adds a common attention layer on top to merge context inconsistencies between input text and image properties. Similarly, Yao et al. [4] developed a multi-modal, multi-interaction, and multi-level neural network using genetic algorithms to extract modality information multiple times to obtain multi-perspective information [5]. Liang et al. [6] proposed a method for determining inconsistency across different modalities by constructing heterogeneous modality graphs and cross-modal graphs for each multi-modal example.

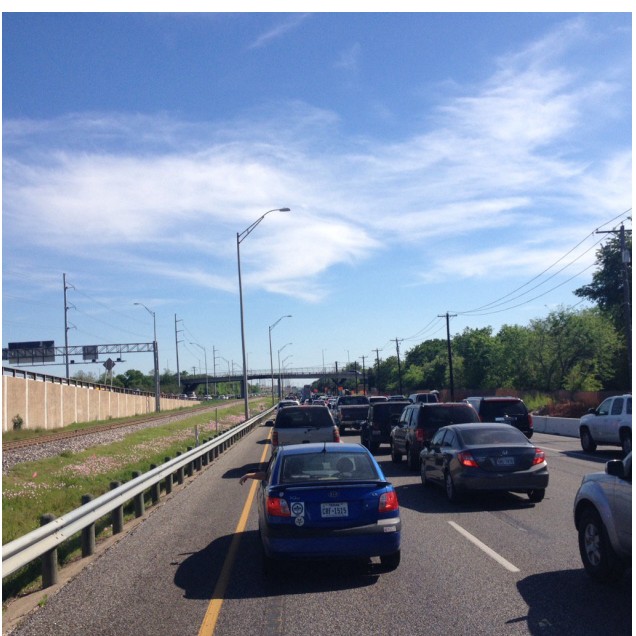

Text: 158 new people a day in # austin # welcome.

**Figure 1.** A multi-modal sarcastic example.

Despite significant progress in multi-modal sarcasm detection, several challenges remain. (1) Achieving a comprehensive understanding of multi-modal sarcasm demands consideration of multiple factors, such as commonsense. However, most existing multi-modal sarcasm detection methods primarily rely on modalities themselves, thereby ignoring the rich semantic information that can be gleaned from the combination of sarcasm samples and external commonsense knowledge. (2) The essence of sarcasm detection lies in measuring emotional inconsistency [7]. Nonetheless, the amount of external commonsense knowledge is substantial, and integrating it with modal semantics for enrichment purposes could either heighten or diminish emotional consistency. Therefore, the effective integration of external commonsense and the sample modality is a crucial issue.

To solve the aforementioned issues, a novel framework is proposed that combines external knowledge and all modalities to improve the recognition of multi-modal sarcasm [8]. For the first issue, a share-transformer [7] is utilized to extract features from each modality. As sarcasm represents a unique way of emotional expression [9], we segment the text using the Natural Language Toolkit (NLTK) [10] tool and then query the emotional polarity score of each word in the text segmentation result using the SenticNet [11] sentiment lexicon. The scores are combined into a vector as the emotional vector. Considering the difficulty in obtaining their emotional tendencies, our method utilizes the sentiment vector corresponding to commonsense as the sentiment vector of the images. For the second issue, we concatenate the emotional information representation obtained by fusing data of each modality with the sentiment vector and perform cross-modal fusion through cross-attention. According to the integrated low-level information in the Fused Layer, the sarcasm is recognized. Our method is evaluated on the Twitter multi-modal irony

dataset and achieves outstanding results, outperforming other models with a weighted precision, recall, and F1 score of 82.11%, 84.77%, and 83.42%, respectively.

Our primary contributions can be succinctly summarized as follows:

- A pioneering model is presented that utilizes an external sentiment [12] lexicon to score modality segmentation and merges the resulting sentiment vectors for multi-modal irony detection;
- A mechanism is proposed for fusing external knowledge with all modalities to reduce noise in inconsistency measurement;
- We perform extensive comparative experiments with other baseline models on the Twitter dataset, and our model outperforms all others, achieving state-of-the-art results.

## 2. Materials and Methods

### 2.1. Related Work

Liu et al. [13] proposed a method of sarcasm detection through atomic-level consistency, utilizing a multi-threaded cross-attention mechanism and article-level consistency based on graph neural networks. Accordingly, it can be observed that the mainstream approach to multi-modal sarcasm detection with deep networks is to extract inconsistent emotional cues from different modalities to discern the true emotions conveyed in multi-modal messages [14]. Li et al. [15] and Veale and Hao [16] emphasized the importance of commonsense knowledge for sarcasm detection. Some instances may include textual information associated with images, and, to address this, Pan et al. [17] and Liang et al. [6] suggested using Optical Character Recognition (OCR) to extract text from images. In more recent work, Liang et al. [18] suggested integrating an object detection framework and labeling information of detected visual objects to address the modality gap. However, the knowledge extracted from these methods may not be expressive enough to convey the information of an image or may be limited to a fixed set, such as nearly a thousand image attributes or ANP classes. Moreover, not every sarcastic post has text overlaid on the image. To overcome these limitations, in this article, we utilize the NLTK [10] tool for text segmentation. We use the text segmentation results and external commonsense knowledge to query the emotional polarity score of each word through the SenticNet [11] sentiment dictionary. Concatenating each modality's data with the sentiment vector results in a representation that fuses emotional information.

### 2.2. Methods

A comprehensive description of our proposed model for multi-modal [19] sarcasm detection is provided in this section shown in Figure 2. The model is composed of 3 primary components: (1) Image Text and Common Modality Representation: This component deploys a visual transformer (ViT) as an image encoder to capture the hidden representation of the image modality. The hidden representation of the common and text modalities is obtained through a bidirectional Long Short Term Memory (LSTM) network. (2) Sentiment Vector [20]: Three emotional words from the image, common, and text are embedded [21] into the original vector processed by the transformer. (3) Attention Fusion module: The attention fusion module fuses the three sentiment vectors that embed the sentiment words. The fusion of text features and common features is input into the fusion module as a query ($Q$) for the multi-head attention operation. Then, image and common feature fusion are used as the keys ($K$) and value ($V$) to adjust the attention to each utterance at any time step.

For model selection, Share-transformer is used to extract features that are then fused through the attention mechanism. Since we need to compare with other baseline models, we chose to use the same DenseNet121 as these to extract image features. In our subsequent work, we will also use other models to extract image features for more in-depth study. The reason why we chose to use SenticNet is because it is capable of extracting more fine-grained sentiment information, which helps in better sarcasm detection.

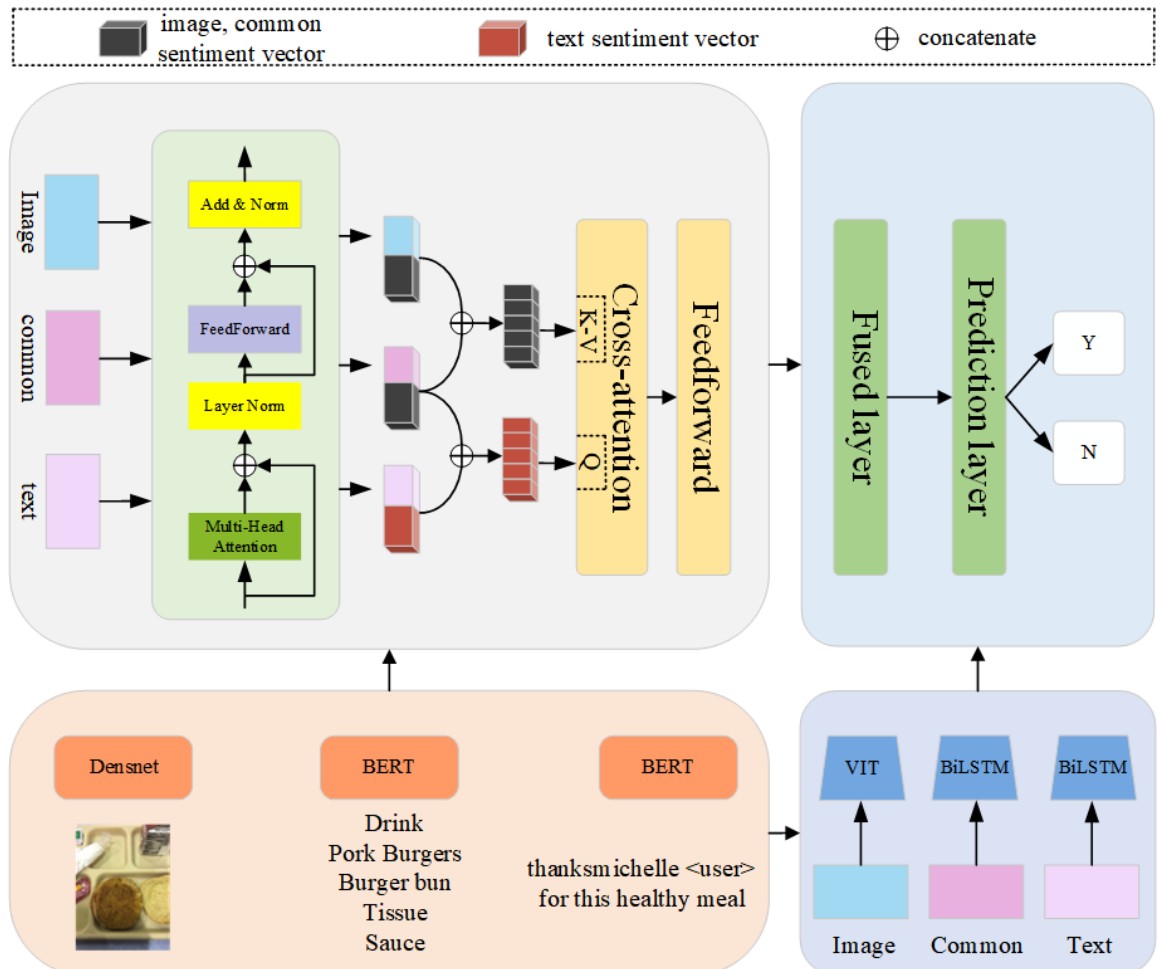

**Figure 2.** Overview of our proposed model. Concatenate the emotional [22] information representation obtained by fusing each modality's data with the sentiment vector to detect sarcasm by integrating low-level information in the Fused Layer.

*2.3. Text Image Modality and Commonsense Representation*

To process image $I$, this paper begins by employing the toolkit proposed by Anderson et al. [23]. This toolkit has been trained to extract a set of bounding boxes representing objects, along with their respective common-object associations. Each visual area $L$ of the bounding box is resized to a size of $244 \times 244$, where both its height ($L_h$) and width ($L_w$) are set to 244. Afterward, in accordance with Dosovitskiy et al. [24], the range $R^{L \times L}$ is converted into a series $I_i = \left\{ p_j \in R^{L/p \times L/p} \right\}_{j=1}^{r}$, where the number of patches is $r = p \times p$.

A bidirectional Long Short-Term Memory (Bi-LSTM) is utilized to generate representations for both the tweet text and commons. The LSTM performs the following operation equation at time step $T$:

$$I_t = \sigma(W_I \cdot X_T + U_I \cdot H_{T-1}) \tag{1}$$

$$F_T = \sigma(W_F \cdot X_T + U_F \cdot H_{T-1}) \tag{2}$$

$$O_T = \sigma(W_O \cdot X_T + U_O \cdot H_{T-1}) \tag{3}$$

$$\tilde{C}_T = \tan H(W_C \cdot X_T + U_C \cdot H_{T-1}) \tag{4}$$

$$C_T = F_T \odot C_{T-1} + I_T \odot \tilde{C}_T \tag{5}$$

$$H_T = O_T \odot \tan H(C_T) \tag{6}$$

Here, $W_I, W_F, W_O, U_I, U_F$, and $U_O$ represent weight matrices; $X_T, H_T$ are the input state and hidden state at time step $T$, respectively; $\sigma$ denotes the sigmoid function; and $\odot$ represents element-wise multiplication.

*2.4. Sentiment Word Embedding (SWE)*

The feature vectors of the three modalities are standardized using the Linear Layer before embedding sentiment words. The resulting vectors are then inputted into the Transformer encoder. In the next stage, taking the example of sentiment word embedding in text mode, every inputted sentence is initially segmented into individual words. Subsequently, sentiment [25] features are extracted through the utilization of the following methodology. For the text mode [26], the text is divided into $n$ words, where the word $c_i$ is queried by SenticNet 7 to obtain the corresponding sentiment score $s_i$. All the sentiment [27] scores form into the vector $(s_1, s_2, \cdots, s_n)$, which is used as the text sentiment vector. Image and commonsense sentiment vectors are obtained in the same process.

*2.5. Multi-Head Attention Fusion Module*

The text, image, and common sentiment vectors that have embedded emotion [28] words (referred to as $F_T, F_I$, and $F_A$) are fed into a unique multi-head attention fusion module. This module aids in the integration of textual information into the model. The self-attention mechanism [29,30] involves the creation of three matrices, namely $Q$ (query), $K$ (key), and $V$ (value), by multiplying the matrix $H$ with weight matrices $(W_Q, W_K, W_V)$. These weight matrices are trained jointly in the self-attention mechanism. Specifically, the matrices $Q$, $K$, and $V$ are calculated as follows: $Q = H \times W^Q$, $K = H \times W^K$, and $V = H \times W^V$. Here, the text sentiment vector $F_T$ is used as the input $Q$ of the fusion module for multi-head attention operation. The image sentiment vector $F_I$ and the common sentiment vector $F_A$ are first fused and then substituted into $K$ and $V$ in order to adjust each time step at any time step in the discourse attention. Thus, every individual modality is transformed into the text vector space, with its corresponding features interconnected. These modalities are then fed into a fully connected layer, which ultimately generates a vector $\in \mathbb{R}^{k \cdot D_T}$ as output. The fusion output of the layer, in addition to the earlier $F_A$ and $F_I$ feature maps, is inputted to the following fusion layer. Then, $m$ multi-head attention fusion layers are stacked together to create a final feature result called $F_{fusion}$, as shown below.

$$F^i_{fusion_1} = \Phi_1\left(F^i_A, F^i_T, F^i_I\right) \tag{7}$$

$$F^i_{fusion_m} = \Phi_m\left(\cdots\left(\Phi_2\left(F^i_A, F^i_{fusion_1}, F^i_I\right)\right)\right), where, i \in [1, k]. \tag{8}$$

In Equations (7) and (8), $\Phi$ denotes the learning function of the multi-head attention fusion module layer. Our fusion strategy differs from previous work utilizing multi-head attention fusion modules while keeping $K$ and $V$ constant. This allows for better modulation of attention between utterances and integration of information across modalities.

The output of the final multi-head attention fusion layer, denoted as $F_{fusion_m}$, is combined with the image and common sentiment vectors ($F_I$ and $F_A$) by concatenation, as shown in the illustration below:

$$F^i_{final} = Concate\left(F^i_{fusion_m}, F^i_A, F^i_I\right) \tag{9}$$

**3. Results**

*3.1. Experimental Settings*

We perform experiments on a benchmark dataset for multi-modal sarcasm detection that is publicly available. Curated by Cai et al. [31] the dataset consists of 24 k samples of tweets, images, and image attributes, containing English-language tweets classified as sarcastic (positive) or non-sarcastic (negative), with corresponding text and images for each tweet. Tweets containing conventional words such as sarcasm, sarcastic, irony, and ironic

were excluded from the dataset. Tweets containing URLs were also excluded to avoid introducing additional information. In addition, we also excluded tweets containing words that often appear alongside sarcastic tweets and therefore may convey irony, such as jokes, humor, and exaggeration. The dataset was divided into a training set, a val set, and a test set in the ratio of 80%:10%:10%. In order to evaluate the model more accurately, we manually checked the development and test sets for labeling accuracy.

To ensure a fair comparison, tweets containing sarcasm as regular words or URLs were removed during data preprocessing. Furthermore, we eliminated tweets that contained words frequently co-occurring with sarcastic tweets that may convey sarcasm (such as jokes, humor, hyperbole, etc.). We used a pre-trained BERT-base model to embed each text-modal word as a 768-dimensional embedding. We employed a pre-trained Densenet model to embed each image region block as a 768-dimensional embedding with $d^T = d^I = 768$. The common dimension is represented by $d^A$, which was set to 512. We adopted accuracy, precision, recall, and F1-score to evaluate the model performance on a dataset constructed by (Cai et al. [31]). The detailed information of the dataset is shown in Table 1. The experimental results were obtained by averaging 10 random initialization runs for our proposed approach.

**Table 1.** The statistics of the multi-modal dataset.

|  | Non Sarcasm | Sarcasm | Total |
| --- | --- | --- | --- |
| Training set | 8642 | 11,174 | 19,816 |
| Val set | 959 | 1451 | 2410 |
| Test set | 959 | 1450 | 2409 |

### 3.2. Comparison Models

Our proposed model was evaluated by comparing it against a range of strongly correlated methods, as outlined below:

(1) Image-modality methods: These models leverage solely visual information for sarcasm detection. For example, **Image** employs ResNet to train a sarcasm classifier, while **ViT** (Dosovitskiy et al. [24]) uses ViTs to identify sarcasm, specifically through the "[class]" token representations;

(2) Text-modality methods: These models rely exclusively on textual information, encompassing **TextCNN** (Kim [32]), a CNN-based deep learning model for text classification; **Bi-LSTM**, a bidirectional LSTM network for classification; **SIARN** (Tay et al. [2]), which employs inner attention to detect sarcasm in the text; **SMSD** (Xiong et al. [33]), which utilizes self-matching networks to capture textual inconsistencies; and **BERT** (Devlin et al. [34]), which accepts "[CLS] text [SEP]" as input for pre-training;

(3) Multi-modal methods: These models utilize both textual and visual information to detect sarcasm. For instance, Cai et al. [31] proposed the **HFM** method, which employs a hierarchical multi-modal feature fusion model for multi-modal sarcasm detection; **Net D&R** (Xu et al. [35]), which employs a decomposition and relation network for cross-modal modeling of modal contrast and semantic association; **Res-BERT** (Pan et al. [17]), which concatenates image features and BERT-based text features to predict sarcasm; **Att-BERT** (Pan et al. [17]), which explores inter-modal attention and co-attention for modeling incongruity in multi-modal sarcasm detection; and **InCross-MGs** (Liang et al. [6]), a graph-based approach that harnesses sarcasm relations from intra- and inter-modal perspectives.

To analyze the effects of various components, we perform ablation studies on multiple variants of the model: (1) *w/o* fusion, which involves fusing information between each modality and only passes a single modality for sarcasm detection; (2) *w/o* emotion, which entails removing sentimental features and setting the edge weight to 1 in the text mode; and (3) *w/o* emotion-fusion, which does not utilize emotion fusion.

### 3.3. Main Results

All baseline models are run over the same training/testing partition as our approach. We utilize Adam as the optimizer with a learning rate of 0.00002 and a mini-batch size of 256. To avoid overfitting, a dropout rate of 0.1 is applied. Table 2 presents the results of our comparison between the text mode, image mode, and text + image mode. Based on our analysis, we have drawn the following conclusions: (1) Our proposed model outperforms the existing baselines on both recall and F1-score metrics, demonstrating its effectiveness in detecting sarcasm in multi-modal content. (2) Statistical significance tests are conducted to compare our model with the baseline model, indicating that our model significantly outperforms the baseline model across the majority of evaluation metrics. (3) Our approach consistently outperforms prior graph-based methods, suggesting that sentiment word embeddings can enhance performance. (4) Text-based methods consistently outperform image-based methods, underscoring the paramount role of textual modality in conveying sarcastic/non-sarcastic information. (5) By integrating information from both image and text modalities, multi-modal approaches surpass their single-modal counterparts, thus proving to be more effective for sarcasm detection across modalities. (6) Our model exhibits superior performance using macro indicators compared to other commonly used indicators. This is indicative of its proficiency in identifying the "negative" category, attributable to an imbalanced category distribution.

**Table 2.** Comparison results of our models with state-of-the-art models according to F1-scores. The highest scores are highlighted in bold.

| Modality | Method | Acc (%) | Pre (%) | Rec (%) | F1 (%) |
|---|---|---|---|---|---|
| image | Image [31] | 64.76 | 54.41 | 70.80 | 61.53 |
| | ViT [24] | 67.83 | 57.93 | 70.07 | 63.43 |
| text | TextCNN [32] | 80.03 | 74.29 | 76.39 | 75.32 |
| | Bi-LSTM | 81.89 | 76.64 | 78.40 | 77.50 |
| | SIARN [2] | 80.55 | 75.56 | 75.68 | 75.61 |
| | SMSD [33] | 80.88 | 76.48 | 75.16 | 75.80 |
| | BERT [34] | 83.83 | 78.70 | 82.26 | 80.21 |
| image + text | HFM [31] | 83.39 | 76.54 | 84.17 | 80.16 |
| | Net D&R [35] | 84.01 | 77.95 | 83.39 | 80.58 |
| | Res-BERT [17] | 84.79 | 77.78 | 84.12 | 80.86 |
| | Att-BERT [17] | 86.04 | 78.61 | 83.28 | 80.92 |
| | InCrossMGs [6] | **86.08** | 81.36 | 84.34 | 82.80 |
| | **Ours** | **86.08** | **82.11** | **84.77** | **83.42** |

### 3.4. Additional Dataset Experiments

In order to verify the validity of the experimental results, we also conducted experiments on another multi-modal sarcasm dataset. This dataset was created by Maity [36] and named 'multibully'. This dataset consists of two modalities, image and text. Each sample is labeled with five tags. We use sarcasm tags. The dataset has a total of 5865 samples. In this the training set and test set are divided in ratio 8:2. The specific experimental results are given in Table 3. Where TextCNN-LSTM extracts text features by adding a recurrent neural network to TextCNN, Maity uses BiGRU and ResNet to extract text and image features respectively. Our method also has high scores in the metrics.

**Table 3.** Experimental results of 'multibully' dataset.

| Modality | Model | Acc (%) | Pre (%) | Rec (%) | F1 (%) | Macro-F1 (%) |
|---|---|---|---|---|---|---|
| Text | TextCNN | 54.22 | 62.14 | 41.17 | 49.53 | 53.92 |
| | TextCNN-LSTM | 57.75 | 59.14 | 53.83 | 56.36 | 57.70 |
| Image | RestNet | 55.66 | 45.53 | 66.97 | 54.21 | 54.57 |
| Text + Image | HFM | 55.43 | 58.50 | 60.97 | 59.71 | 54.92 |
| | Maity [36] | 58.70 | 64.61 | 56.37 | 60.12 | 58.64 |
| | **Ours** | **59.51** | 61.67 | **63.03** | **62.34** | **59.31** |

## 4. Discussion

Previous studies on detecting sarcasm have primarily concentrated on textual discourse data (Zhang et al. [37]; Tay et al. [2]; Babanejad et al. [38]). However, multi-modal sarcasm detection aims to recognize sarcastic expressions across various modalities, as opposed to just text-based sarcasm detection. To address multi-modal-based sarcasm detection, Cai et al. [31] proposed predicting five attributes for each image using a pre-trained ResNet model (He et al. [39]) as the third modality for sarcasm detection. Pan et al. [17] introduced cross-modal attention and co-attention mechanisms to understand the contradictory nature of sarcasm. Liang et al. [6] used a heterogeneous graph structure for graph-based methods to capture sarcasm characteristics from both intra- and inter-modal perspectives. Nevertheless, their approaches strive to capture the visual information of the entire image while disregarding the emotional expressions between diverse modalities. In contrast to previous approaches, our novel framework fuses all modalities with external knowledge to facilitate multi-modal sarcasm recognition.

### 4.1. Ablation Study

We conducted an ablation study to assess the impact of the various components of our proposed model. The results, presented in Table 4, show that removing the fusion module ($w/o$ fusion) significantly diminishes the performance, highlighting the importance of fusion. Further, the removal of the emotion-fusion component ($w/o$ emotion-fusion) leads to a considerable decline in performance, underscoring the significance of incorporating emotion-fusion into the overall model fusion module. Based on the ($w/o$ emotion) experiment outcomes, it can be concluded that the performance of the model for sarcasm detection is also drastically reduced by removing all the sentiment vectors proposed in this paper.

**Table 4.** Experimental results of ablation study.

| | Acc (%) | Pre (%) | Rec (%) | F1 (%) |
|---|---|---|---|---|
| **Ours** | **86.08** | **82.11** | **84.77** | **83.42** |
| $w/o$ fusion | 80.35 | 79.63 | 80.65 | 80.14 |
| $w/o$ emotion-fusion | 81.32 | 80.52 | 82.42 | 81.46 |
| $w/o$ emotion | 81.68 | 79.88 | 84.67 | 82.21 |

### 4.2. Multi-Modal Experiements Analysis

We conducted an experimental analysis with different modalities to evaluate the impact of the different modal parts of the proposed model. The results, shown in Table 5, indicate that there is a significant decrease in the performance of verifying whether the model is satirical or not only through the image modality, which is due to the ambiguity of the information expressed in the image modality. In contrast, only a small decrease in performance is seen when validation is performed only through the text modality; this is because, from the text, it is more straightforward to derive what the writer is trying to say. However, since the work completed is to detect the sarcasm of the tweets, it is necessary

to compare the meaning expressed by the textual modality with the image modality in order to truly identify whether the meaning is sarcastic or not. The experimental data also show the highest performance scores of the model after modal fusion, which highlights the importance of modal fusion.

**Table 5.** The result of multi-modal experiments.

| Modality | Acc (%) | Pre (%) | Rec (%) | F1 (%) |
|---|---|---|---|---|
| Text | 83.36 | 80.09 | 79.46 | 79.78 |
| Image | 72.44 | 67.24 | 64.93 | 66.06 |
| Text + Image | **86.08** | **82.11** | **84.77** | **83.42** |

*4.3. Case Study*

Figure 3 displays sample tweets that our model accurately detects as sarcastic. The experimental results demonstrate that sentiment word embeddings facilitate precise sarcasm detection. For instance, in Figure 3a, the text "find the donut photo (not mine) . . .enjoy" contrasts with the "sad grey expression" in the accompanying image. The sentimental feature attention mechanism prioritizes the word "enjoy" in the text, which is the critical element in the image common, alongside detecting the gesture "rolling eyes". Consequently, our model can identify the tweet as sarcastic. Similarly, for the sarcasm detection in Figure 3b, the text ""i'm 6 ′ 3" . . . "great" legroom in # united economy . . . " contrasts with the "confined space" in the accompanying image. With the help of sentiment vectors, our model learns the inconsistent dependencies of the two modalities and predicts the correct outcome for these examples.

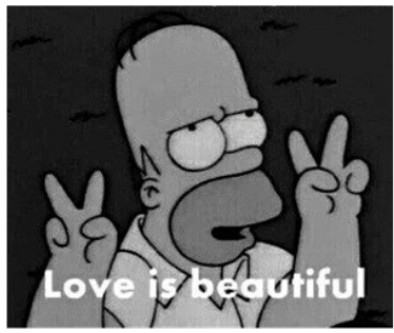

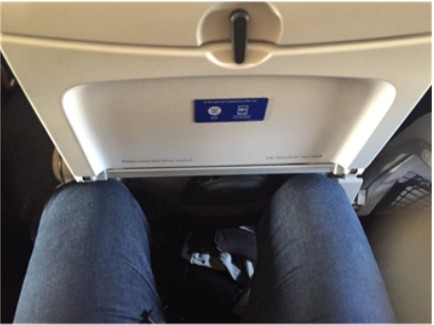

(a) Common: grey , rolling eyes , open mouth , the Simpsons
Text: found donut photo (not mine) .. enjoy

(b) Common: great , legroom in , united economy
Text: ' i \'m 6 \' 3 " ... " great " legroom in # united economy ... '

**Figure 3.** Sarcasticexamples of case study.

*4.4. Error Analysis*

We conducted an error analysis on our experimental results and discovered that the vast majority of blunders can be attributed to samples with significant textual information on images. An example is presented in Figure 4. Our observation of Figure 4a reveals that it solely comprises textual information, thus presenting a challenge for our model. Since our model lacks external image attributes, it struggles to process the intricate interplay between the textual content and the image morphologies. Likewise, in the image presented in Figure 4b, only a hazy backdrop is discernible, rendering it an arduous task to establish any connection between the information conveyed by the image and its textual content. We contend that identifying sarcasm in certain words of the accompanying text, amidst ambiguous imagery, poses a formidable challenge not just for deep learning models but even for human annotators.

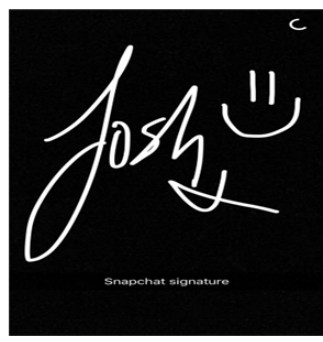

(a) took me so long to get this guys , i waited for hours . thanks <user>

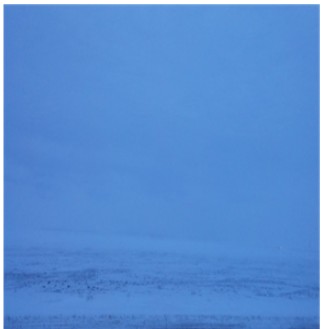

(b) visibility on hwy 8 is awesome ! <user>

**Figure 4.** Sarcastic examples of error analysis.

## 5. Conclusions

This study proposes a novel model architecture for multi-modal sarcasm detection, where the essence is to embed sentiment words into feature vectors of three different modalities to enhance the capture of incongruous emotions expressed by sarcasm. In contrast to previous research that solely considers modality, our work extends this approach by incorporating sentimental features into sarcasm detection. Our experimental results reveal a significant improvement in sarcasm detection accuracy by embedding emotional words into feature vectors. The assessment is carried out extensively on public benchmark datasets, demonstrating that our proposed method surpasses current state-of-the-art baseline methods. In daily life, people sometimes express the opposite of what they think in their minds, which forms sarcasm. In future research, we intend to incorporate psychological commonsense knowledge into text and investigate sarcasm in image-generated text.

**Author Contributions:** Conceptualization, H.F., H.L. and D.J.; Methodology, H.F. and H.L.; Software, H.F. and H.L.; Validation, H.F. and H.L.; Formal analysis, H.F.; Investigation, H.F.; Resources, H.F.; Data curation, H.F.; Writing—original draft, H.F.; Writing—review & editing, H.F., H.L. and L.X.; Visualization, H.F.; Supervision, H.F. and D.J.; Project administration, H.F., J.L. and D.J.; Funding acquisition, H.W. and D.J. All authors have read and agreed to the published version of the manuscript.

**Funding:** This research was funded by Science and Technology Major Project of Guangdong Province: STKJ2021005, STKJ202209002, STKJ2023076; National Natural Science Foundation of China: 62372283, 62206163; Natural Science Foundation of Guangdong Province: 2019A1515010943; Opening Project of Guangdong Province Key Laboratory of Information Security Technology: 2020B1212060078.

**Data Availability Statement:** The data presented in this study are available in the public domain: https://github.com/headacheboy/data-of-multimodal-sarcasm-detection, accessed on 1 January 2024.

**Acknowledgments:** The authors would like to acknowledge and thank all reviewers for their constructive and helpful reviews.

**Conflicts of Interest:** Hongling Wang was employed by the company Shenzen Tobo Software Co., Ltd. The remaining authors declare that the research was conducted in the absence of any commercial or financial relationships that could be construed as a potential conflict of interest.

## Abbreviations

The following abbreviations are used in this manuscript:

| | |
|---|---|
| NLTK | Natural Language Toolkit |
| ViT | Visual Transformer |
| LSTM | Long Short-Term Memory |
| Bi-LSTM | Bidirectional Long Short-Term Memory |
| SWE | Sentiment Word Embedding |

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
