# Peer review of "Multi-Modal Sarcasm Detection with Sentiment Word Embedding"

_electronics, doi:10.3390/electronics13050855_

Round 1

Reviewer 1 Report

Comments and Suggestions for Authors

The authors work in the very interesting and challenging area of sarcasm detection. They focus on twitter tweet type of content and propose a new multimodal approach that considers both image and text. The proposed novelty is that they also consider a sentiment dictionary.

The paper is well written, in the sense that the scope is well defined, the approach is described to sufficient detail, the presented experiments seem to provide a fair evaluation and the claimed conclusions stem easily from the experimental results.

The value of the contribution can be summarized in Table 2, in which the proposed approach is shown to outperform a wide range of other methods, particularly the ones that are also bi-modal.

Author Response

Dear Sir/Madam,

Thank you very much for all your comments. We made substantial changes in response to your comments, and fell the quality of this paper had a great improvement as a result of updating this paper taking your suggestions. Thank you very much for your valuable comments and please check our updates in response to each comment:

Point 1: The authors work in the very interesting and challenging area of sarcasm detection. They focus on twitter tweet type of content and propose a new multimodal approach that considers both image and text. The proposed novelty is that they also consider a sentiment dictionary.

Response 1: Thank you for recognizing our work. We will continue to conduct more in-depth research in our future work.

Point 2: The paper is well written, in the sense that the scope is well defined, the approach is described to sufficient detail, the presented experiments seem to provide a fair evaluation and the claimed conclusions stem easily from the experimental results.

Response 2: Thank you for recognizing our work. We will continue to conduct more in-depth research in our future work.

Point 3: The value of the contribution can be summarized in Table 2, in which the proposed approach is shown to outperform a wide range of other methods, particularly the ones that are also bi-modal.

Response 3: Thank you for recognizing our work. We will continue to conduct more in-depth research in our future work.

Reviewer 2 Report

Comments and Suggestions for Authors

1. In Lines 160-161, where does the testing multimodal dataset come from? Did the authors collect it by some API? Or did the authors obtain this dataset from previous studies?

2. Where is the ground truth data confirming whether the test/training/val tweet is sarcastic?

3. Table 2 says the "image+text" approaches typically outperform the "image" or "text" alone approaches in detecting sarcasm in multimodal content. However, the authors did not mention the availability/accessibility of such multimodal content. For example, what is the percentage of "image+text" content available compared to the whole Tweet dataset during a specific period? The authors might need more clarification on this validity issue.

4. The authors might need more examples/discussion in 4.3. Case Study. I don't think Figure 3(b) is a sarcastic example; it is a more neutral post. The author might need to 1. clarify this detection procedure (I know you elaborated on it in 2. Materials and Methods. Since it is called a "Case Study," you should explain how your low-level information was fused in your cross-modal fusion layer led to such sarcasm detection) 2. or use more examples.

Author Response

Dear Sir/Madam,

Thank you very much for all your comments. We made substantial changes in response to your comments, and fell the quality of this paper had a great improvement as a result of updating this paper taking your suggestions. Thank you very much for your valuable comments and please check our updates in response to each comment:

Response to reviewer 2:

Point 1: In Lines 160-161, where does the testing multimodal dataset come from? Did the authors collect it by some API? Or did the authors obtain this dataset from previous studies?

Response 1: Thanks for your suggestion. The multimodal dataset we tested is from the paper "Multi-Modal Sarcasm Detection in Twitter with Hierarchical Fusion Model" from Cai et al. In the paper we also add a specific description of it.

The changes are reflected in lines 171 to 181.

Point 2: Where is the ground truth data confirming whether the test/training/val tweet is sarcastic?

Response 2: Thanks for your suggestion. The dataset was divided into a training set, a val set, and a test set in the ratio of 80%:10%:10%. Ironically there are 11,174 pieces of data in the training set, 1,451 in the val set, and 1,450 in the test set. The training set without irony had 8,642 entries, the val set had 959 entries, and the test set had 959 entries.These data are presented in Table 1.

The changes are reflected in lines 171 to 181.

Point 3: Table 2 says the "image+text" approaches typically outperform the "image" or "text" alone approaches in detecting sarcasm in multimodal content. However, the authors did not mention the availability/accessibility of such multimodal content. For example, what is the percentage of "image+text" content available compared to the whole Tweet dataset during a specific period? The authors might need more clarification on this validity issue.

Response 3: Thanks for your suggestion. For the purposes of this article, the percentage of "image + text" content available compared to the entire Tweet dataset during a specific period is 80 percent.

The changes are reflected in lines 171 to 181.

Point 4:The authors might need more examples/discussion in 4.3. Case Study. I don't think Figure 3(b) is a sarcastic example; it is a more neutral post. The author might need to 1. clarify this detection procedure (I know you elaborated on it in 2. Materials and Methods. Since it is called a "Case Study," you should explain how your low-level information was fused in your cross-modal fusion layer led to such sarcasm detection) 2. or use more examples.

Response 4: Thanks for your suggestion. We chose a more ironic post to replace Figure b in the case study to better show that our model, with the help of sentiment vectors, can learn the incoherent dependencies of the two modalities and predict the correct outcome for these examples.

The changes are reflected in lines 290 to 294 and figure 3.

Reviewer 3 Report

Comments and Suggestions for Authors

This paper proposes an innovative multimodal sarcasm detection model that splits and scores different modalities using an external sentiment lexicon and merges sentiment vectors. The method has a certain innovation, and experimental results have also proved that its effect is better than multiple baseline models. However, there is still room for improvement in the model and paper:

  1. The algorithm description lacks details to fully reproduce the sentiment word embedding module. It is recommended that the authors further supplement the reasons for choosing SenticNet, the calculation method of sentiment scores, etc., which will enable readers to more fully understand and reproduce this module.
  2. The paper does not provide theoretical basis and analysis for model selection. The Share-transformer, DenseNet and sentiment lexicon SenticNet used in this paper are directly cited from existing work without explaining why these module combinations are chosen. It is recommended to increase relevant proof.
  3. In the experimental part, in order to prove the effectiveness of sentiment word embedding, contrast experiments with and without this module need to be designed. In the current group analysis (Table 3), the "w/o emotion" corresponding experiment only removes the sentiment weights in the text, and cannot well reflect the gain of the sentiment word embedding module. It is recommended to supplement direct comparisons with and without this module.
  4. In the evaluation metrics of sarcasm detection, in addition to accuracy, recall, F1 values, human consistency metrics can be introduced to evaluate the correlation between model and human judgment. This helps to intuitively understand the algorithm performance.
Comments on the Quality of English Language

Minor editing of English language required

Author Response

Dear Sir/Madam,

Thank you very much for all your comments. We made substantial changes in response to your comments, and fell the quality of this paper had a great improvement as a result of updating this paper taking your suggestions. Thank you very much for your valuable comments and please check our updates in response to each comment:

Response to reviewer 3:

Point 1:The algorithm description lacks details to fully reproduce the sentiment word embedding module. It is recommended that the authors further supplement the reasons for choosing SenticNet, the calculation method of sentiment scores, etc., which will enable readers to more fully understand and reproduce this module.

Response 1: Thanks for your suggestion. The reason why we chose to use SenticNet is because it is capable of extracting more fine-grained sentiment information, which helps in better sarcasm detection.

The changes are reflected in lines 109 to 115.

Point 2:The paper does not provide theoretical basis and analysis for model selection. The Share-transformer, DenseNet and sentiment lexicon SenticNet used in this paper are directly cited from existing work without explaining why these module combinations are chosen. It is recommended to increase relevant proof.

Response 2: Thanks for your suggestion. For model selection we explain here, where Share-transformer is used to extract features which are then fused through the attention mechanism. Since we need to compare with other baseline models, we chose to use the same DenseNet as them to extract image features. In our subsequent work, we will also use other models to extract image features for more in-depth study. The reason why we chose to use SenticNet is because it is capable of extracting more fine-grained sentiment information, which helps in better sarcasm detection.

The changes are reflected in lines 109 to 115.

Point 3:In the experimental part, in order to prove the effectiveness of sentiment word embedding, contrast experiments with and without this module need to be designed. In the current group analysis (Table 3), the "w/o emotion" corresponding experiment only removes the sentiment weights in the text, and cannot well reflect the gain of the sentiment word embedding module. It is recommended to supplement direct comparisons with and without this module.

Response 3: Thanks for your suggestion. We apologize for any confusion in your reading due to the lack of clarity in our writing. In Table 3, the "w/o emotion " corresponds to what should be the experiment with all emotion vectors removed. We have made the correction in the text.

The changes are reflected in lines 266 to 269.

Point 4: In the evaluation metrics of sarcasm detection, in addition to accuracy, recall, F1 values, human consistency metrics can be introduced to evaluate the correlation between model and human judgment. This helps to intuitively understand the algorithm performance.

Response 4: Thanks for your suggestion. In order to make a fair comparison with other models, we chose to use the same four evaluation metrics for scoring model performance. We are very sorry that we did not introduce a human consistency metrics to evaluate the correlation between the model and human judgment in this work. We will include this metric in future work for deeper investigation.

Reviewer 4 Report

Comments and Suggestions for Authors

The paper presents a method for sarcasm detection using sentiment-word embeeding. The method utilize lexicon to score modality segmentation and merges the resulting sentiment vectors for achieving multi-modality. The contribution of the work itself seems limited as it consists in incorporating sentiment based on a external resource, the remaining of the method is quite standard. The introduction of the paper needs to highlight the novelty of the approach. More importantly, the paper lacks of an in-depth discussion of related literature. This is fundamental to contextualize the work regarding related ones in an subject that is being intensively investigated.

The experimental results are in general poorly described. First, results corresponds to experiments with a single datasets, an additional dataset would be valuable to validate the results. Also, there is not report of the statistical significance of the differences achieved in each case. The methodology of experiments is not mentioned, for example how the dataset was split and which parameters were used for the different algorithms. The baselines choose for comparison need to be justified as they are not designed to incorporate sentiment, so the comparison may be unfair.

Although the paper is in general understandable, there are many typos in the paper that need to be corrected.

Comments on the Quality of English Language

English writing is in general understandable, there are minor mistakes and typos.

Author Response

Dear Sir/Madam,

Thank you very much for all your comments. We made substantial changes in response to your comments, and fell the quality of this paper had a great improvement as a result of updating this paper taking your suggestions. Thank you very much for your valuable comments and please check our updates in response to each comment:

Response to reviewer 4:

Point 1:  The paper presents a method for sarcasm detection using sentiment-word embeeding. The method utilize lexicon to score modality segmentation and merges the resulting sentiment vectors for achieving multi-modality. The contribution of the work itself seems limited as it consists in incorporating sentiment based on a external resource, the remaining of the method is quite standard. The introduction of the paper needs to highlight the novelty of the approach. More importantly, the paper lacks of an in-depth discussion of related literature. This is fundamental to contextualize the work regarding related ones in an subject that is being intensively investigated.

Response 1: Thanks for your suggestion. We have added a description of the method in the introduction section and discussed of related literature in more depth.

The changes are reflected in lines 47 to 60.

Point 2: The experimental results are in general poorly described. First, results corresponds to experiments with a single datasets, an additional dataset would be valuable to validate the results. Also, there is not report of the statistical significance of the differences achieved in each case. The methodology of experiments is not mentioned, for example how the dataset was split and which parameters were used for the different algorithms. The baselines choose for comparison need to be justified as they are not designed to incorporate sentiment, so the comparison may be unfair.

Response 2: Thanks for your suggestion. We added a multimodal satirical dataset for experiments to verify the validity of the results. This dataset consists of two modalities: image and text. Each sample is labeled with 5 labels. We use satirical labels. There are 5865 samples in this dataset. In which the training set and test set are divided in ratio 8:2. To ensure fairness, we chose the same selected baseline as the previous method for comparison. In this way, we verify the improvement of multimodal sarcasm detection performance after sentiment word embedding.

The changes are reflected in lines 237 to 245 and table 3.

Point 3: Although the paper is in general understandable, there are many typos in the paper that need to be corrected.

Response 3: Thanks for your suggestion. We checked the paper and corrected any typos found in it.

Round 2

Reviewer 2 Report

Comments and Suggestions for Authors

The authors have properly addressed my comments. I wish them the best of luck with their publication and research.

Author Response

Dear Sir/Madam,

Thank you very much for all your comments. We made substantial changes in response to your comments, and fell the quality of this paper had a great improvement as a result of updating this paper taking your suggestions. Thank you very much for your valuable comments and please check our updates in response to each comment:

Response to reviewer 2:

Point 1: The authors have properly addressed my comments. I wish them the best of luck with their publication and research.

Response 1: Thank you for recognizing our work. We will continue to conduct more in-depth research in our future work.

Reviewer 3 Report

Comments and Suggestions for Authors

The authors successfully addressed the issues. It can be published in the current format.

Author Response

Dear Sir/Madam,

Thank you very much for all your comments. We made substantial changes in response to your comments, and fell the quality of this paper had a great improvement as a result of updating this paper taking your suggestions. Thank you very much for your valuable comments and please check our updates in response to each comment:

Response to reviewer 3:

Point 1: The authors successfully addressed the issues. It can be published in the current format.

Response 1: Thank you for recognizing our work. We will continue to conduct more in-depth research in our future work.

Reviewer 4 Report

Comments and Suggestions for Authors

Most of the concerns indicated in the revision were addressed by the authors, there are still some improvements that can be made. First, although the paper includes now some new references to related works, they have been included in the introduction. A separate Related Works section needs to be included to discussed the state-of-the-art in the subject. This should include the advantages and disadvantages of current approaches to contextualize the contribution of this work. Second, in the experimental results the method is compared with several baselines. More details of how this comparison was done would help to clarify some points. For example, the baselines were run over the same partitions of training/test? Which parameters were used for the algorithms? there was some optimization of the parameters? What about the statistical significance of the results?

There are types throughout the paper to be corrected. Mostly, all references lack of an space or added a comma, as the following. “Intentions[1]” -> “Intentions [1]” and “Gupta et al.,[3]” -> “Gupta et al. [3]”.

Comments on the Quality of English Language

The English writing is understandable.

Author Response

Dear Sir/Madam,

Thank you very much for all your comments. We made substantial changes in response to your comments, and fell the quality of this paper had a great improvement as a result of updating this paper taking your suggestions. Thank you very much for your valuable comments and please check our updates in response to each comment:

Response to reviewer 4:

Point 1: First, although the paper includes now some new references to related works, they have been included in the introduction. A separate Related Works section needs to be included to discussed the state-of-the-art in the subject. This should include the advantages and disadvantages of current approaches to contextualize the contribution of this work. 

Response 1: Thanks for your suggestion. We have added a chapter on "Related Works" in section 2.1 of the article, in which we describe the strengths and weaknesses of the current approach and the contributions of our work.

The changes are reflected in lines 78 to 97.

Point 2: Second, in the experimental results the method is compared with several baselines. More details of how this comparison was done would help to clarify some points. For example, the baselines were run over the same partitions of training/test? Which parameters were used for the algorithms? there was some optimization of the parameters? What about the statistical significance of the results?

Response 2: Thanks for your suggestion. Our experiments were run on the same training/testing partition as all baseline models. For the parameters used and the optimization of the parameters, we also provide additional descriptions in section 3.1 and section 3.3.

The changes are reflected in lines 185 to 195 and lines 224 to 241.

Point 3: There are types throughout the paper to be corrected. Mostly, all references lack of an space or added a comma, as the following. “Intentions[1]” -> “Intentions [1]” and “Gupta et al.,[3]” -> “Gupta et al. [3]”.

Response 3: Thanks for your suggestion. We have corrected the formatting of all references in the text.